# Enhancing the Treatment of Uncontrolled Inflammation through the Targeted Delivery of TPCA-1-Loaded Nanoparticles

**DOI:** 10.3390/pharmaceutics15102435

**Published:** 2023-10-09

**Authors:** Zhaozhao Chen, Lu Tang, Lili Luo, Wenjing Luo, Yingying Li, Xindi Wang, Linlin Huang, Yu Hu, Heng Mei

**Affiliations:** 1Institute of Hematology, Union Hospital, Tongji Medical College, Huazhong University of Science and Technology, Wuhan 430022, China; zhaozhaochen@hust.edu.cn (Z.C.); lu_tang@hust.edu.cn (L.T.); d202281890@hust.edu.cn (L.L.); weerfi1211@hust.edu.cn (W.L.); yingyli@hust.edu.cn (Y.L.); xindiwang@hust.edu.cn (X.W.); linlinhuang@hust.edu.cn (L.H.); 2Hubei Clinical Medical Center of Cell Therapy for Neoplastic Disease, Wuhan 430022, China; 3Key Laboratory of Biological Targeted Therapy of Hubei Province, Wuhan 430022, China

**Keywords:** targeted delivery, nanoparticles, PECAM-1, TPCA-1, uncontrolled inflammation

## Abstract

Uncontrolled inflammation is a pathological state that underlies many diseases. Despite the development of numerous anti-inflammatory agents, the treatment of uncontrolled inflammation remains a challenging task. We developed a targeted delivery system for [5-(p-fluorophenyl)-2-ureido]thiophene-3-carboxamide (TPCA-1), a potent inhibitor of the NF-κB signaling pathway. The system comprises TPCA-1-loaded nanoparticles (NPs) functionalized with a monoclonal antibody (mAb) that specifically binds to the break point of the IgD6 region of the platelet/endothelial cell adhesion molecule-1 (PECAM-1) extracellular segment that is overexposed on the injured endothelium and activated macrophages during the pathogenesis of inflammation. In vitro binding and cellular uptake experiments revealed that the mAb modification on NPs could significantly enhance uptake by both Raw264.7 and HUVEC compared with unmodified NPs. In studies conducted at the cellular level focusing on anti-inflammatory and antioxidant effects, this formulation was found to effectively inhibit M1 polarization of macrophages, downregulate the secretion of pro-inflammatory cytokines, and reduce the production of reactive oxygen species (ROS) and nitric oxide (NO). In an animal model of vascular endothelial injury with acute inflammation, these NPs were capable of delivering TPCA-1 to inflammatory lesions in a targeted manner. Compared with the free agent-treated group, the NP-treated group exhibited reduced infiltration of inflammatory cells. In conclusion, our study demonstrates that this targeted delivery of TPCA-1-loaded NPs represents a promising strategy for improved mitigation of uncontrolled inflammation.

## 1. Introduction

Inflammation is a natural defense response in the body under pathological conditions, but excessive inflammatory reactions can cause health challenges [1,2]. For instance, COVID-19 can induce immune dysregulation that culminates in an unbridled state of hyperinflammation, which might facilitate the development of acute respiratory distress syndrome (ARDS) or multiorgan dysfunction syndrome (MODS), ultimately resulting in the mortality of affected individuals [3,4]. At present, the mainstream clinical treatment for uncontrolled inflammation includes the administration of glucocorticoids and tocilizumab [5,6,7]. Nevertheless, overdose glucocorticoid administration is highly likely to induce osteonecrosis [8], which has been observed in COVID-19 patients [9]. Moreover, tocilizumab has also raised concerns about adverse drug reactions, such as lung and liver sarcoidosis [10]. Therefore, further efforts are needed to develop targeted delivery systems with low toxicity to mitigate inflammatory responses. Growing evidence has indicated that endothelial injury and macrophage activation play important roles in the development of hyperinflammation [11,12,13,14,15]. First, PECAM-1 is a glycoprotein constitutively expressed on endothelial cells and most leukocyte subtypes, including macrophages and T cells, to varying degrees [16,17,18]. As a transmembrane molecule, the extracellular segment of PECAM-1 consists of six Ig homeodomains [19], and upon inflammatory stimulation, the expression of PECAM-1 is upregulated [20]. In addition, the IgD1–IgD5 regions of the extracellular segment are cleaved, and the IgD6 region is exposed, releasing soluble PECAM-1 (sPECAM-1) [21,22]. In addition, the intense inflammatory responses in injured areas are driven by the recruitment of resident and circulatory macrophages, further amplifying the inflammatory cascade [23]. Finally, it is equally important that during the progression of inflammation, leukocyte interaction with the endothelium leads to vessel wall damage, enlarged endothelial intercellular space, and enhanced leakage, offering a unique opportunity for the targeted delivery and accumulation of nanoparticles at the site of the lesion through enhanced permeability and retention (EPR) effects [24,25,26].

Previous investigations have employed drug delivery platforms including gold nanocages [27], hybrid liposomes–nanovesicles [28], and platelet-derived extracellular vesicles [29] for delivering TPCA-1 to treat inflammation-related diseases. Nevertheless, poor biodegradability will affect the physiological clearance of gold-based NPs in vivo [30,31]. Furthermore, the complex preparation process of hybrid nanovesicles and extracellular vesicles has raised technical challenges, and there have been large heterogeneities in the final formulations [32]. Hydroxyethyl starch (HES) is a modified starch derivative formed by introducing hydroxyethyl groups onto starch molecules through the use of epichlorohydrin under alkaline conditions. It is commonly employed in clinical practice as a plasma substitute for volume expansion in shock patients [33]. Due to its exceptional water solubility, biocompatibility, degradability, and favorable surface modification characteristics, HES has frequently been investigated as a nanocarrier for the delivery of diagnostic and therapeutic agents to malignant tumors [34]. Cholesterol (CH), on the other hand, is an endogenous lipid molecule in metabolic processes with inherent hydrophobic and lipophilic properties. Self-assembly of amphiphilic polymers at the oil–water interface is a classical approach for nanoparticle construction [29,30]. Given the clinical utility of HES and the presence of CH within the human body, amphiphilic polymers formed by chemically conjugating HES and CH [35], along with the resulting self-assembled nanoparticles, would exhibit excellent biocompatibility and clinical acceptability. 

Hence, based on the collective performance of PECAM-1 in endotheliocytes and macrophages, targeted delivery of NPs targeting PECAM-1 or its extracellular segment cleavage points may improve the treatment of acute inflammation. In this study, an active targeted drug delivery system (mAb-TPCA-1@HCNPs) was prepared by exploiting a monoclonal antibody specific for the IgD6 region cleavage points of PECAM-1 as the targeting moiety, NPs based on an amphiphilic polymer (HES-CH) as the drug carrier, and the hydrophobic anti-inflammatory agent TPCA-1 as the model drug. The mAb-TPCA-1@HCNPs were also subjected to comprehensive characterization. To assess their targeting capabilities, we conducted in vitro binding studies and cellular uptake experiments using fluorescence-labeled NPs, as well as in vivo biodistribution investigations. Furthermore, we investigated the anti-inflammatory and antioxidant properties of this novel formulation, both at the cellular level and in a murine model of vascular endothelial injury with acute inflammation induced by lipopolysaccharide (LPS) (Figure 1).

## 2. Materials and Methods

### 2.1. Materials

HES 40/0.5 with an average molecular weight (Mw) of 40 kDa and a hydroxyethyl substitution degree of 50% was purchased from Wuhan HUST Life Sci & Tech Co., Ltd. (Wuhan, China). Cholesterol, succinic anhydride, coumarin-6 (C6), N,N′-disuccinimide carbonate (DSC), LPS (0111: B4), 1-hydroxybenzotriazole hydrate (HOBt), 1-(3-dimethylaminopropyl)-3-ethylcarbodiimide hydrochloride (EDC-HCl), type Ⅰ collagenase, and DNase were ordered from Sigma (New York, NY, USA). DiR iodide was obtained from AmyJet Scientific (Wuhan, China). TPCA-1 was obtained from MedChemExpress (Monmouth Junction, NJ, USA). PECAM-1 mAb (PECAM-1-4G6, targeted PECAM-1 extracellular segment D6 domain, 150kD) was a gift from Wisconsin Blood Center (Milwaukee, WI, USA). The CCK8 kit was purchased from Dojindo (Kyushu, Japan). The mouse/human PECAM-1 ELISA kit was purchased from Elabscience (Wuhan, China). Mouse TNF-α and IL-6 ELISA kits were from MultiSciences (Hangzhou, China). The ROS detection kit, DAF-FMDA NO fluorescent probe detection kit, and lysosomal red fluorescent probe were ordered from Beyotime Biotechnology (Shanghai, China). PE-anti-CD80, FITC-anti-CD45, APC-anti-CD3, APC-anti-F4/80, and PE-anti-CD11b were purchased from BioLegend (San Diego, CA, USA). Fixable Viability Stain 780 was purchased from BD (Franklin Lakes, NJ, USA). Anti-MPO, anti-CD68, anti-CD45, and anti-CD3 were purchased from ABclonal (Wuhan, China). The human umbilical vein endothelial cell (HUVEC) line, mouse aortic endothelial cell (MAEC) line, and RAW264.7 cell line were obtained from the American Type Culture Collection. They were cultured according to the recommended suggestions. Female BalB/c mice (six to eight weeks old) were purchased from Beijing Vital River Laboratory Animal Technology Co., Ltd. (Beijing, China). All experiments were carried out according to the regulations and standards of the ethics committee of Huazhong University of Science and Technology (approval number: 3135).

### 2.2. Preparation of mAb-TPCA-1@HCNPs

The amphiphilic hydroxyethyl starch-coupled cholesterol polymer HES-CH was synthesized as previously reported [35]. NPs were fabricated via the pickering emulsion solvent evaporation method [36,37,38]. Briefly, 50 mg of HES-CH was dissolved in 50 mL of deionized water to obtain a 1 mg/mL aqueous solution by a probe sonicator (Qsonica, Newtown, CT, USA) for 10 min (100 W). Then, 5 mL of TPCA-1 solution (1 mg/mL, dissolved in chloroform) was slowly added to the above system. Simultaneously, the probe sonicator was opened for 5 min to obtain a homogeneous oil/water-mixed solution. After removing chloroform with a RE-52A rotary evaporator (Yarong Biochemical Instrument Factory, Shanghai, China), NP solution was obtained by centrifugation at 5000 rpm for 10 min (Eppendorf, Germany), and the precipitate was discarded. The supernatant was dialyzed overnight to remove unloaded TPCA-1 and subsequently lyophilized to obtain TPCA-1@HCNP powder. Similarly, the fluorescently labeled tracer NPs, DiR@HCNPs and C6@HCNPs, were prepared as described above. The preparation workflows are shown in Figure 2.

At room temperature, sufficient DSC was added to a 1 mg/mL TPCA-1@HCNP aqueous solution, and the mixture was stirred overnight to obtain a sufficient amidation reaction, followed by dialysis (MWCO, 10,000 kDa) for 24 h. Then, 25 μg PECAM-1 4G6 was added, the mixture was stirred for 3 h at room temperature, and the free antibody was removed by centrifugation (13,000 rpm, 10 min) to obtain mAb-TPCA-1@HCNPs. Similarly, mAb-DiR@HCNPs and mAb-C6@HCNPs were prepared as described above.

### 2.3. Characterization of mAb-TPCA-1@HCNPs

The drug loading capacity (DLC) and encapsulation efficiency (EE) were also investigated. Gradient concentrations of TPCA-1 DMSO solution, 200 µg/mL, 100 µg/mL, 20 µg/mL, 10 µg/mL, and 1 µg/mL, were prepared and determined by high-performance liquid chromatography (HPLC) to obtain a standard curve. Afterward, DMSO was used to fracture the TPCA-1@HCNPs (incubation for 30 min, and the concentration of NPs was 100 µg/mL), and the obtained samples were detected by HPLC to obtain the concentration and mass of TPCA-1. The DLC and EE were calculated using Equations (1) and (2), respectively:DLC = (Wt(TPCA-1))/(Wt(TPCA-1@HCNPs)) × 100%(1)
EE = (Wt(TPCA-1))/(Wt(Total TPCA-1)) × 100%(2)
where Wt (TPCA-1) is the mass of TPCA-1 in TPCA-1@HCNPs, Wt (TPCA-1@HCNPs) is the mass of TPCA-1@HCNPs nanoparticles, and Wt (Total TPCA-1) is the total mass of TPCA-1 invested in the preparation of nanoparticles.

The conjugation efficiency of mAbs to the TPCA-1-loaded NPs was also evaluated. Briefly, a secondary antibody, Alexa Fluor 488-conjugated goat anti-mouse IgG (H+L), was added to the aqueous solution of mAb-TPCA-1@HCNPs and reacted on the shaker at room temperature for 1 h. Then, the solution obtained was centrifuged at 13,000 rpm for 10 min, and the supernatant was discarded. The precipitate was subsequently washed three times and resuspended with ddH_2_O. Finally, the NPs solution was diluted to 1:10,000 and detected by nano-flow cytometry (NanoFCM, Xiamen, China).

The drug release efficiencies of TPCA-1@HCNPs and mAb-TPCA@HCNPs at variable pH (5.3, 6.5, and 7.4) were also performed. In brief, the solution of TPCA-1@HCNPs or mAb-TPCA@HCNPs was added into a dialysis bag (MWCO, 50 kDa). Then the dialysis bag was immersed in the release medium, which consisted of ddH_2_O containing 0.05% Tween at different pH conditions. This system was subsequently placed on a constant-temperature shaker at 100 rpm/min and 37 °C. At determined time points, 1 mL of solution was obtained outside the bag, and an equal amount of release medium was added. The TPCA-1 concentration in the solution was measured by the HPLC system. The mobile phase was a mixture of acetonitrile and water (CH_3_CN:H_2_O = 30:70, *v*/*v*) with a flow rate of 1.0 mL/min. The sample injection volume was 20 µL, and the detector wavelength was 270 nm.

The particle size distribution and zeta potential of NPs were measured using a Malvern Zetasizer ZEN3600 Nano ZS (Malvern, UK). The morphology of mAb-TPCA-1@HCNPs was observed using transmission electron microscopy (TEM) (H-7000FA, Hitachi, Japan) after placing a drop of NP solution on the copper mesh and staining with 0.2% (*w*/*w*) phosphotungstic acid for 60 s. The stability was investigated by monitoring particle size during seven days of storage in PBS, 10% FBS, and RPMI 1640 at 4 °C.

### 2.4. Hemolytic Assay

A hemolytic assay was performed to investigate the blood compatibility of mAb-TPCA-1@HCNPs. Red blood cells were obtained from BALB/c mice. Samples incubated with PBS as the negative control served as the positive control, and samples incubated with 1% Triton served as the positive control. Subsequently, each sample was centrifuged at 1500 rpm for 15 min. The absorbance of the supernatant was detected at 540 nm using a microplate reader. The hemolysis ratio was calculated using Equation (3) as follows:HR = (Odt − ODn)/(Odp − ODn) × 100%(3)
where ODt is the optical density (OD) value of each experimental group, ODn is the OD value of the negative control group, and ODp is the OD value of the positive control group.

### 2.5. Cell Culture

RAW264.7 cells (mouse leukemia cells of monocyte macrophages) and HUVECs were obtained from ATCC. The cells were cultured in Dulbecco’s modified Eagle’s medium (DMEM)/high-glucose culture medium solution containing 10% fetal bovine serum (FBS) and 1% penicillin/streptomycin in a humidified incubator at 37 °C under 5% CO_2_.

### 2.6. In Vivo and In Vitro Validation of PECAM-1 Extracellular Segment Cleavage under Inflammatory Stimulation

For in vitro validation, HUVECs and RAW264.7 cells were plated at 2 × 10^5^ cells in six-well plates and placed in an incubator overnight for adherence. The next day, 100 ng/mL LPS was added and incubated for 24 h. Then, the supernatant was obtained, followed by centrifugation at 1000× *g* for 20 min, and the concentration of sPECAM-1 was measured by ELISA. For in vivo validation, BALB/c mice were intraperitoneally injected with LPS at a dose of 7.5 mg/kg, and control groups were injected with an equivalent volume of PBS. After 24 h, the mice were sacrificed, and then serum and lung tissue samples were obtained. The concentration of sPECAM-1 was measured by ELISA, and the presence of the extracellular segment cleavage of PECAM-1 was detected by immunofluorescence.

### 2.7. In Vitro Cell Cytotoxicity Assay

We evaluated the cytotoxicity of mAb-TPCA-1@HCNPs to HUVECs and RAW264.7 cells by CCK-8 assay. The cells were seeded in ninety-six-well plates, treated with various concentrations of NPs for 24 h, and then incubated with CCK-8 for 1 h at 37 °C. Finally, the OD values were measured at 450 nm by a microplate reader.

### 2.8. Cytokine Assay

For ex vivo stimulation, RAW264.7 cells were seeded and pretreated with 100 ng/mL LPS for 1 h, and then free TPCA-1 and mAb-TPCA-1@HCNPs were added to each group. After 24 h, cytokine levels in the cell supernatant were measured by ELISA.

### 2.9. Intracellular ROS and NO Measurement

ROS and NO production were evaluated using ROS and NO assay kits, respectively. To stimulate cells, RAW264.7 cells were pretreated with 100 ng/mL LPS for 1 h. Subsequently, free TPCA-1 and mAb-TPCA-1@HCNPs were added to each group for 24 h. Each group was then cultured with 10 µM DCFH-DA or DAF-FM DA for 20 min at 37 °C. Subsequently, the abovementioned cells were washed three times with PBS. Then, the cells were collected, and the change in intracellular ROS or NO levels was evaluated using flow cytometry.

### 2.10. Macrophage Polarization Analysis

RAW264.7 cells were treated with 100 ng/mL LPS for 1 h in advance to simulate macrophages under inflammatory conditions, followed by the addition of TPCA-1 and mAb-TPCA-1@HCNPs to each group for another 24 h. Cells were collected in Eppendorf tubes and resuspended as single cells. Then, 1 μL of FcR blocking reagent was added to each tube and incubated at 4 °C for 10 min. Subsequently, 1 μL of PE-anti-CD80 was added to each tube and cultured at room temperature for 30 min. Finally, after three washes, the stained cells were analyzed by flow cytometry (BD, Franklin Lakes, NJ, USA) to evaluate the polarization of macrophages.

### 2.11. In Vitro Binding and Cellular Uptake Experiments

To stimulate macrophages under inflammatory conditions, RAW264.7 cells and HUVECs were prestimulated with 100 ng/mL LPS for 24 h. For binding and cellular uptake experiments, stimulated cells were coincubated with mAb-C6@HCNPs for different time intervals, and the dose of coumarin-6 was 100 ng/mL. At predetermined times, the cells were washed three times with PBS to remove free NPs, stained with Lysotracker Red, fixed with 4% paraformaldehyde, and then stained with DAPI. The cells were washed with PBS three times, and eventually, binding and cellular uptake were observed using Dragonfly/CR-DFLY-201-40 turntable laser confocal microscopy (Andor Technology, Belfast, UK). Additionally, we further quantified the fluorescence intensity using flow cytometry (BD, Franklin Lakes, NJ, USA).

### 2.12. In Vivo Pharmacokinetics Study of NPs in Mice

We substituted DiR for TPCA-1 to evaluate the pharmacokinetic profile of mAb-TPCA-1@HCNPs after intravenous (i.v.) injection in mice, and the dose of DiR was 2 mg/kg. At predetermined time points, an equivalent volume of peripheral blood samples was collected from the retro-orbital vein and placed in ninety-six-well plates. Subsequently, the mean fluorescence intensity (MFI) of the samples was determined using a Lago X Imaging System (Spectral Instruments Imaging, Tucson, AZ, USA).

### 2.13. Animal Model Establishment and Treatment

A mouse model of acute uncontrolled inflammation was induced according to a previous study [39]. Briefly, BALB/c mice were anesthetized and then injected intraperitoneally with LPS (7.5 mg/kg). After treatment with LPS, mice were administered PBS, HCNPs, free TPCA-1, TPCA-1@HCNPs, and mAb-TPCA-1@HCNPs. The mice were sacrificed after 24 h.

### 2.14. In Vivo Targeting and Biodistribution of NPs in a Mouse Model

We created the mouse model as mentioned above. After 2 h, mice were intravenously injected with mAb-TPCA-1@HCNPs (*n* = 4), and the dose of DiR was 2 mg/kg. In addition, healthy mice and model mice intravenously injected with free DiR and isotype IgG-DiR@HCNPs were used as control groups. After 24 h, mice were euthanized, and their organs were collected. Subsequently, the fluorescence intensity of the organs was detected on a Lago X Imaging System (Spectral Instruments Imaging, USA). For the histological study, model mice were intravenously injected with mAb-C6@HCNPs to label NPs. Then, we collected the lungs, fixed them in 4% paraformaldehyde, and embedded them in paraffin. After a series of workflows, we used a CD68 primary antibody followed by cyanine 5-conjugated secondary antibody incubation to detect macrophages. All slides were analyzed using a fluorescence microscope (Olmypus, Tokyo, Japan).

### 2.15. In Vivo Therapeutic Efficacy Experiment

Mice were randomly divided into six groups as follows: a healthy control group, LPS+PBS, LPS+HCNPs, LPS+free TPCA-1, LPS+isotype IgG-TPCA-1@HCNPs, and LPS+mAb-TPCA-1@HCNPs, in which the dose of TPCA-1 was 1 mg/kg. We constructed the sepsis ALI model by intraperitoneal injection of LPS (7.5 mg/kg) as mentioned above, followed by intervention with the above drugs. After 24 h, the mice were sacrificed, and the lungs were collected and washed with PBS three times. The lung tissues were cut up and homogenized with type Ⅰ collagenase and DNase, and then the obtained homogenate was filtered through a 40 µm strainer three times to acquire single-cell suspensions. Subsequently, red blood cell lysis buffer was added to the suspensions and incubated for 40 min on ice. After being washed three times with PBS, the cell suspensions were incubated with 1 µL of FcR blocking reagent for 10 min at 4 °C. Next, the samples were stained with Fixable Viability Stain 780 and various fluorescent antibodies. The cells were analyzed using flow cytometry (BD, Franklin Lakes, NJ, USA).

### 2.16. Detection of ROS Levels in Mouse Lung Tissue

Lung tissues were collected, fixed in 4% paraformaldehyde, and embedded in paraffin. To obtain paraffin sections, samples were cut into sections with a thickness of 4 mm. After being deparaffinized in water, the sections were stained with dihydroethidium (DHE) and incubated at 37 °C for 40 min. Then, the DHE dye was removed, and the cells were washed with PBS three times. Subsequently, the sections were incubated with DAPI and washed with PBS. Finally, the slides were sealed and observed using a fluorescence microscope.

### 2.17. Biosafety Evaluation of NPs in Mice

Healthy mice were divided into two groups: the experimental group was intravenously injected with mAb-TPCA-1@HCNPs, and the other group was intravenously injected with an equal volume of PBS (*n* = 5). We monitored the weight of the mice every two days for 14 days. After two weeks, the mice were euthanized, and their main organs (liver, spleen, kidneys, heart, and lungs) were collected and washed three times, fixed in 4% paraformaldehyde, and embedded in paraffin. The HE staining of organs was performed according to standard procedures. For comparison of lung tissue damage scores, the scoring criteria were based on a previous study [40,41]. In short, four pathological indicators were scored on a scale of 0 to 4: (1) alveolar congestion, (2) hemorrhage, (3) leukocyte infiltration or neutrophil aggregation in the airspace or vessel walls, and (4) alveolar wall thickness. A score of 0 represented no lung injury, and scores of 1, 2, 3, and 4 indicated mild, moderate, severe, and very severe lung injury, respectively. The sum of all scores was recorded for comparative analysis. Simultaneously, peripheral blood was collected from mice by removing the eyeball, and we used an auto hematology analyzer (Genrui, Shenzhen, China) to examine the routine blood metrics, including red blood cell (RBC) count, white blood cell (WBC) count, platelet (PLT) count, hematocrit (HCT), hemoglobin (HGB) levels, mean corpuscular volume (MCV), mean corpuscular hemoglobin (MCH), and mean corpuscular hemoglobin concentration (MCHC). Moreover, for the blood biochemical test, the whole blood sample was added to the procoagulant tube and placed at room temperature for 2 h. Then, after centrifugation (3000 rpm, 15 min) at 4 °C, the plasma was collected, and a biochemical analyzer (Biobase, Jinan, China) was used to measure the levels of indicators including serum albumin (ALB), alanine transaminase (ALT), aspartate aminotransferase (AST), blood urea nitrogen (BUN), serum creatinine (CREA), and urine creatinine (UREA).

### 2.18. Statistical Analysis

All experimental data are presented as the mean ± standard deviation (SD) (*n* = 3–5). A one-way analysis of variance was performed to determine the statistical significance of differences among groups. For multiple comparisons, Tukey’s post hoc test for significant differences was used. All statistical analyses were performed using GraphPrism (v8.0.2). Significance in the figures is indicated by asterisks: * *p* < 0.05, ** *p* < 0.01, ****p* < 0.005, **** *p* < 0.001.

## 3. Results

### 3.1. Preparation and Characterization of mAb-TPCA-1@HCNPs

To develop a targeted drug delivery platform, a mAb specific for the IgD6 region fracture points of PECAM-1, which are usually overexpressed on endotheliocytes and macrophages [20,21,22], was selected as the targeting moiety. As shown in Figure 1A,B, the TEM images exhibited that TPCA-1@HCNPs and mAb-TPCA-1@HCNPs were of a regular and spherical morphology, and their size was approximately 100 nm. In addition to showing the size distribution, dynamic light scattering (DLS) (Figure 1C) further demonstrated that the diameter of NPs increased slightly with the encapsulation of TPCA-1 and the modification of mAb. Both HCNPs and TPCA-1@HCNPs were positively charged. The encapsulation of TPCA-1 did not affect the zeta potential of the nanoparticles. Once modified with mAb, the zeta potential of mAb-TPCA-1@HCNPs decreased to 19.367 ± 0.379 mV (Figure 1D) because mAb as a protein was negatively charged. These results were consistent with a previous study, which also demonstrated that antibody modification slightly affected the size and zeta potential of polymer NPs [42,43,44]. The stability of mAb-TPCA-1@HCNPs was investigated based on the particle size within a week, as shown in Figure 1E and Appendix A. Within seven days, the particle size varied, but no significant particle size changes were observed.

We also determined the conjugation efficiency of mAbs to the TPCA-1-loaded NPs by nano-flow cytometry. Compared to the unmodified and blank NPs, we found that up to 98.03 ± 0.4256% mAbs were coupled onto TPCA-1@HCNPs (Figure 1G and Appendix A). The in vitro release kinetics of TPCA-1 from both nanotherapeutics in variable pH conditions displayed a comparable biphasic profile (Figure 1H–J and Appendix A), underscoring that the mAb modification had a negligible impact on the in vitro release dynamics of TPCA-1 from the NPs. 

In many studies, NPs were intravenously injected into the blood vessels and would interact with various blood cells directly. Hemolysis assays are usually utilized to evaluate the toxicity and biosafety of biomaterials [45,46]. Even at a concentration of 500 μg/mL, the hemolysis ratio was lower than 2%, and the structure and morphology of erythrocytes were not changed. However, the most positive control, 1% Triton X-100, lysed all the red blood cells (Figure 1F, Appendix A).

### 3.2. In Vivo and In Vitro Validation of PECAM-1 Extracellular Segment Cleavage

The extracellular segment cleavage of PECAM-1 in endotheliocytes and macrophages is illustrated in Figure 2A. As shown in Figure 2B, compared to that in the control group, the concentration of sPECAM-1 was elevated 1.5-fold when HUVECs were exposed to inflammatory stimulation. Similarly, this phenomenon also occurred in RAW264.7 cells. Once stimulated with LPS, the level of sPECAM-1 in the LPS group was 101.8 ± 9.803 pg/mL, which was much higher than the 17.71 ± 2.453 pg/mL level in the control group (Figure 2C). In addition, we measured the level of sPECAM-1 in a mouse sepsis ALI model, which was 838.7 ± 77.71 pg/mL and 6.8-fold that of the healthy control group (120 ± 50.32 pg/mL) (Figure 2D). In addition, the upregulated expression of the extracellular segment cleavage points of PECAM-1 in the lung tissues of the ALI model was also verified by immunofluorescence (Figure 2E). These results indicated that the inflammatory environment has a great impact on the cleavage of PECAM-1 in vitro and in vivo. Reportedly, as an immune response to stress, the IgD1–IgD5 region of the extracellular segment was cleaved, and sPECAM-1 was consequently released not only in an ex vivo stimulation model [20,47] but also in inflammation-related diseases in vivo [21,48]. Our research data are consistent with the abovementioned findings, confirming the collective response of PECAM-1 in endotheliocytes and macrophages, which is the theoretical basis of the targeted nanotherapeutics-based drug delivery system to improve the treatment of ALI.

### 3.3. In Vitro Cell Cytotoxicity and Therapeutic Effect

Hyperinflammation in ALI is characterized by severe endothelial injury and uncontrollable inflammatory cell activation in the lungs, consequently inducing acute respiratory distress syndrome (ARDS) and systemic inflammatory response syndrome (SIRS) [49,50,51]. We hypothesized that mAb-TPCA-1@HCNPs would elicit attenuation effects on inflammation in vitro.

In this study, we chose TPCA-1 as the model drug encapsulated in NPs, which is an inhibitor of the NF-κB signaling pathway [43,44] and suppresses LPS-induced human monocyte production of TNF-α, IL-6, and IL-8 [43,45]. A previous study reported that TPCA-1 could alleviate the inflammatory response in patients with pneumonia [46]. First, we evaluated the in vitro cytotoxicity of mAb-TPCA-1@HCNPs in HUVECs and RAW264.7 cell lines (Figure 3A,B), and we determined the IC50 of these two cell lines, respectively (Appendix A). Then, we studied the immunomodulatory effect of mAb-TPCA-1@HCNPs on activated macrophages in an inflammatory environment. The recruitment of resident and circulatory alveolar macrophages plays a driving role in contributing to the extensive and excessive inflammatory responses in compromised lungs [23,52]. During the inflammatory response, there is a dynamic conversion between immune-stimulative M1 and immunosuppressive M2 phenotypes in the polarization of macrophages [48]. Once the immune response is initiated, resident and circulatory alveolar macrophages undergo a phenotypic transition from M2 to M1 [23,53]. The increase in the number of proinflammatory M1 macrophages results in the elevated secretion of proinflammatory cytokines such as IL-6 and TNF-α and the production of ROS and NO, which mediate injury to lung tissues and further amplify the inflammatory network [54,55,56]. To mimic the in vitro inflammatory microenvironment, we activated RAW264.7 cells using LPS (100 ng/mL). As indicated in Figure 3C,D and Appendix A, after treatment with TPCA-1, TPCA-1@HCNPs, IgG-TPCA-1@HCNPs, and mAb-TPCA-1@HCNPs, the RAW264.7 cells exhibited a lower proportion of CD80+, and as expected, the macrophages in the LPS- and LPS+HCNP-treated groups displayed a typical M1 phenotype. Moreover, all treatments with TPCA-1 reduced the secretion of IL-6 (Figure 3E) and TNF-α (Figure 3F), as well as the production of intracellular ROS (Figure 3G) and NO (Appendix A).

The above results could be explained by the successful loading of TPCA-1, which allows the NP-based delivery system to have immunoregulatory and anti-inflammatory functions.

### 3.4. In Vitro Binding and Cellular Uptake Experiments

We then investigated the targeting capacity and cellular uptake to demonstrate the targeting specificity and endocytosis mechanism of mAb-TPCA-1@HCNPs. To visualize the nanoparticles, we substituted fluorescein C6 for TPCA-1. After challenge with LPS, activated macrophages were incubated with mAb-C6@HCNPs for 2 h. Using confocal laser scanning microscopy, it was observed that mAb-C6@HCNPs showed a higher affinity toward activated macrophages compared with the nonactivated macrophages and the group blocked by excessive monoclonal antibodies (Figure 4A and Appendix A). The binding effect was similarly revealed in challenged endotheliocytes (Figure 4B and Appendix A), which are indispensable to the pathophysiology of acute inflammation [57]. This targeting ability was significantly weakened in resting endothelial cells and in the blocked group. These favorable results are attributed to the cellular segment cleavage of PECAM-1 in inflammatory conditions [21]. The above clues indicated that mAb-TPCA-1@HCNPs were bispecific for macrophages and endotheliocytes under inflammatory conditions in vitro.

Cellular internalization and endocytic pathways of nanoparticles are crucial to the delivery efficiency and bioavailability of nanocarriers. In our study, the macrophage cell line RAW264.7 was cocultured with mAb-C6@HCNPs for 0.5, 4, 8, 12, and 24 h, and the RAW264.7 cells were then stained with Lysotracker, a bioprobe for lysosomes with red fluorescence emission, which was utilized to ascertain the colocalization of the C6-labeled nanocarriers. At the indicated time points, we monitored the dynamic endocytosis behavior of mAb-C6@HCNPs. We observed the fluorescence changes of mAb-C6@HCNPs and lysosome signal colocalization, demonstrating the cellular uptake of nanoparticles via a classical endocytosis pathway and the escape of the drugs from lysosomes (Figure 4C and Appendix A). We found that the cellular uptake efficiency occurred in a time-dependent manner and that mAb-C6@HCNPs showed maximum colocalization with lysosomes in 8 h, in which the Pearson’s R value peaked at 0.8356, and after this time point, the value started to decrease (Figure 4C), suggesting that mAb-TPCA-1@HCNPs initiated escape from lysosomes after 8 h, when the nanoparticles were cocultured with RAW264.7 cells. In addition, the concentration-dependent cellular uptake of mAb-C6@HCNPs was also evaluated. As shown in Appendix A, we found that the cellular uptake efficiency occurred in a time-dependent manner, as expected. Collectively, these results indicated the excellent targeted binding efficiency and cellular internalization of mAb-TPCA-1@HCNPs.

### 3.5. In Vivo Pharmacokinetics and Biodistribution Study

The in vivo pharmacokinetic profile of mAb-TPCA-1@HCNPs was first studied in healthy mice. To study the pharmacokinetics of the nanoparticles, we substituted fluorescein DiR for TPCA-1 to label the NPs. After intravenous injection of DiR@HCNPs, fluorescence imaging of whole blood collected from mice suggested that DiR@HCNPs gradually degraded and were almost completely cleared from the blood after 48 h (Figure 5A,B). The in vivo pharmacokinetic profile is consistent with mAb-TPCA-1@HCNPs (Figure 5C,D). Hence, the intravenously injected nanotherapeutics delivery system could be present in normal mice for 48 h, and a very small amount of drug remained after 2 days, demonstrating that it has a favorable long-term circulation effect in peripheral blood. We then investigated the targeting capacity and evaluated the ability of mAb-TPCA-1@HCNPs to accumulate in various organs in a mouse model of LPS-induced sepsis ALI (Appendix A). Here, the model mice were intravenously injected with mAb-DiR@HCNPs, and free DiR and isotype IgG-DiR@HCNPs (at the same dose of DiR) were intravenously injected as control groups. As expected, we observed that mAb-DiR@HCNPs in the inflamed lungs of sepsis ALI model mice demonstrated the strongest fluorescence intensity compared with those in the lungs of healthy mice or isotype IgG-DiR@HCNPs and free dye-treated ALI model mice (Figure 5E,F). Similarly, the fluorescence intensity of mAb-DiR@HCNPs in the liver and spleen was also superior to that of the other groups (Figure 5E,F). Furthermore, we labeled nanoparticles with coumarin-6 for immunofluorescence to track the biodistribution of mAb-TPCA-1@HCNPs. The immunofluorescence of lung tissue also showed preferable accumulation of mAb-C6@HCNPs compared with free coumarin-6 and isotype IgG-C6@HCNP treatment or in the lungs of healthy mice (Figure 5G), confirming the targeting capacity of mAb-TPCA-1@HCNPs at the acute lung inflammation site.

Our tracking and biodistribution results of NPs are in accordance with a previous study that used an endotoxemia model induced by LPS [39]. The favorable target capacity of mAb-TPCA-1@HCNPs is attributed to the following: Endothelial injury and loss of integrity are hallmarks of sepsis, thus promoting capillary leakage [58], which facilitates nanoparticle accumulation at the site of lung lesions through EPR effects [24]. Additionally, the extracellular segment cleavage of PECAM-1 in endotheliocytes and macrophages in model mice is considered an active target of mAb-TPCA-1@HCNPs.

### 3.6. In Vivo Therapeutic Efficacy Experiment

Encouraged by the above experimental results, we further evaluated the therapeutic effect of mAb-TPCA-1@HCNPs in vivo. To demonstrate the therapeutic efficacy of targeted delivery, we established a sepsis model of ALI by challenging mice with LPS (7.5 mg/kg) [39] (Appendix A). After two hours, PBS, HCNPs, free TPCA-1 (1 mg/kg), IgG-TPCA-1@HCNPs, and mAb-TPCA-1@HCNPs (equal to 1 mg/kg TPCA-1) were intravenously administered. All mice were sacrificed 24 h after intravenous injection for analysis. First, histological examinations were carried out to demonstrate that LPS-induced ALI model mice exhibited damaged alveolar structures, increased alveolar wall thickness and interstitial exudation of alveoli, alveolar congestion and hemorrhage, and infiltration of inflammatory cells in the airspace or vessel walls (Figure 6A,D). This result was consistent with that of the LPS+HCNP-treated group, suggesting that as a nanocarrier, there was no therapeutic efficacy of HCNPs. However, the free TPCA-1 drug showed slight alleviation of lung injury, and IgG-TPCA-1@HCNPs exhibited moderately reduced lesions in lung tissues. Moreover, mAb-TPCA-1@HCNPs demonstrated a significantly higher therapeutic effect in the model mice, in which the alveolar structures of lung tissues were remarkably recovered, and lung edema and the infiltration of inflammatory cells were obviously reduced (Figure 6A,D). ROS plays a crucial role in the maintenance of cellular processes and functions in the body. The excessive generation of ROS under the pathological conditions of sepsis and ALI leads to increased endothelial permeability and lung injury [59,60]. DHE staining of lung tissues was also performed to detect ROS generation. As shown in Appendix A, mAb-TPCA-1@HCNP treatment showed a significant ROS-scavenging capability during ALI, while free TPCA-1 treatment slightly suppressed the generation of ROS. During the pathogenesis of ALI, circulatory and resident macrophages are activated and shift to the M1 phenotype [23], and excessive activated T-cell cytotoxicity also partly contributes to both a dysfunctional immune response and unrestrained immunopathology [61]. We detected the infiltration of macrophages and T cells using flow cytometry. As shown in Figure 6B,E and Appendix A, compared with mice treated with free TPCA-1 drugs and IgG-TPCA-1@HCNPs, the level of CD45+CD3+ T cells obviously decreased. In addition, the infiltration of CD45+CD11b+F4/80+ macrophages was also relieved (Figure 6C,F and Appendix A). Furthermore, the immunofluorescence of lung tissues visualized the infiltration of inflammatory cells and demonstrated the superior anti-inflammatory capacity of mAb-TPCA-1@HCNPs compared with other treatments (Appendix A).

### 3.7. Biosafety Evaluation in Mice

We also systemically evaluated the biocompatibility and toxicity of mAb-TPCA-1@HCNPs. Two groups of BAL/Bc mice were intravenously injected with PBS and mAb-TPCA-1@HCNPs at a high dose of 100 mg/kg, and mice were monitored for 14 days. We first tested routine blood and blood biochemistry indicators. The routine blood metrics (RBC count, WBC count, PLT count, HCT, HGB levels, MCV, MCH, and MCHC) (Figure 7A–H), the levels of liver function biomarkers ALB, ALT, and AST (Figure 7I–K), and the levels of kidney function biomarkers BUN, CREA, and UREA (Figure 7L–N) were comparable to those of the PBS-treated group and within normal ranges. In addition, as shown in Appendix A, there was no obvious tissue damage in mice treated with mAb-TPCA-1@HCNPs. Similarly, during the monitoring period, the body weight of the mice in the two groups slightly increased (Figure 7O), suggesting no significant systemic toxicity of our nanotherapeutics.

Overall, these preliminary data confirm that mAb-TPCA-1@HCNPs are of favorable biosafety and biocompatibility in vivo.

## 4. Discussion

Inflammatory diseases encompass a wide spectrum of conditions, ranging from acute injuries to chronic disorders. Effective drug delivery is a key factor in the successful management of these conditions, particularly uncontrollable inflammation. Various drug delivery systems have been explored to enhance therapeutic outcomes while minimizing side effects. Notably, these strategies include liposomes, nanoparticles, micelles, and hydrogels [62,63], each with its own advantages and challenges. While liposomes and nanoparticles offer improved drug solubility and bioavailability, micelles and hydrogels provide sustained release profiles [62,64]. In our study, we opted for HES-CH NPs as the chosen vehicle for TPCA-1 due to their modifiability, biocompatibility, and ability to be tailored for targeted delivery. PECAM-1 plays a pivotal role in vascular biology and inflammation. However, PECAM-1's role in inflammation is multifaceted and context-dependent. While it is primarily associated with anti-inflammatory functions such as leukocyte transmigration and endothelial cell signaling, it can also mediate proinflammatory responses under certain conditions [16,18]. It is expressed on the surface of endothelial cells and macrophages [18]. When stimulated by inflammation, the extracellular segment of PECAM-1 would be shed, exposing the IgD6 region breakpoint [21]. This makes it a prime candidate for targeted drug delivery to inflamed tissues.

Our developed drug delivery system capitalizes on the precisely targeted delivery of anti-inflammatory agents for hyperinflammatory lesions. To begin with, the NPs in our platform are designed to specifically recognize the extracellular IgD6 region breakpoint of PECAM-1 overexpressed on inflamed endothelial cells and activated macrophages. This selective binding ensures that the drug payload is delivered precisely to the site of inflammation. In addition, as passive targeted delivery, the “EPR effect” caused by vascular injury in acute inflammation further enhances the accumulation of drugs in the lesion. Last but not least, TPCA-1, as a potent NF-κB inhibitor, disrupts the inflammatory signaling cascade within immune cells, especially macrophages. This inhibition leads to a reduction in the infiltration of pro-inflammatory immune cells, ultimately attenuating the inflammatory response and tissue injury.

## 5. Conclusions

In summary, we successfully designed a targeted drug delivery system modified with mAbs of high affinity for the extracellular segment cleavage points of PECAM-1 in endotheliocytes and macrophages as a targeting moiety that could not only improve the delivery of inflammatory lesions but also specifically interfere with the activation and conversion of the main contributors to acute inflammation. We found that mAb-TPCA-1@HCNPs could accumulate at the site of inflammatory lung lesions and facilitate the delivery of anti-inflammatory agents by intravenous administration. An in vivo therapeutic study revealed that targeted mAb-TPCA-1@HCNPs demonstrated superior anti-inflammatory therapeutic efficacy compared with that of free TPCA-1 in a mouse model of uncontrolled inflammation. In summary, this is the first preclinical study in which a nanoplatform-based targeted strategy was applied to improve the treatment of uncontrolled inflammation. This work may also offer insights into treatments for patients with severe COVID-19.

## Data Availability

The data presented in this study are available in Appendix A: https://www.mdpi.com/article/10.3390/pharmaceutics15102435/s1.

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
