# Peer review of "Enhancing the Treatment of Uncontrolled Inflammation through the Targeted Delivery of TPCA-1-Loaded Nanoparticles"

_pharmaceutics, 2023, doi:10.3390/pharmaceutics15102435_

Round 1

Reviewer 1 Report

In this report, Chen et al., designed and synthesized a dual-targeting delivery system for [5-(p-fluorophenyl)-2-ureido] thiophene-3-carboxamide (TPCA-1), a potent inhibitor of the NF-κB signaling pathway. Also, they showed vascular endothelial injury with acute inflammation, these NPs were capable of delivering TPCA-1 to inflammatory lesions in a targeted manner. However, the authors need to address the following concerns.

1.        The abstract is not written well and doesn’t reflect the complete results of the study. Can be rewritten?

2.        The drug release efficiencies of TPCA-1@HCNPs, and mAb-TPCA@HCNPs at variable pH were not performed and discussed.

3.        Are there any changes observed in blood profiles of mice observed post administration of formulations? Did the formulation cause any signs of distress and aberrations in their growth patterns?

4.        The authors have not determined the IC50 values of the formulation. The inclusion of this data would provide more information about the dose administration.

5.        Did authors perform any biocompatible studies with the formulations?

6.        What proportion do authors agree on this current approach would fit with existing standard therapies?

7.        The manuscript can be revised further for grammatical and typological errors.

       The manuscript can be revised further for grammatical and typological errors.

Reviewer 2 Report

In this manuscript, the authors assessed the effectiveness of using TPCA-1-loaded NP to improve the treatment of uncontrolled inflammation. I believe the findings herein reported will bring a strong contribution to the understanding of uncontrolled inflammation related to some severe diseases. I therefore recommend acceptance after the following minor revisions:

1. Can the authors provide direct evidence of the successful conjugation of monoclonal antibody to the drug-loaded NP? In other words, what was the conjugation efficiency?

2.  In Fig, 5E, please label each line with the corresponding organ (Lung, liver, etc.)

English language is fine. Minor editing (Typos) is required.

Reviewer 3 Report

Abstract and title. Please, rephrase the term «dual targeting» since it is mainly used for nanoparticles with two or more targeting moieties. Since authors used only one targeting moiety, it could confuse readers.

Introduction. 

Please, exclude conclusions from the introduction (rephrase or exclude the last sentence).

Enlarge the part dedicated to the choice of drug delivery platform. Why did the authors choose this type of nanoparticle? Please, describe the advantages and compare them with other investigations of developed targeted TPCA-1-loaded nanoparticles, e.g., gold nanoparticles, to highlight the novelty.

Materials and methods. 

What is the molecular weight of used monoclonal antibodies? 

Please, add detailed information about cell culture conditions (medium, supplements). 

Results Please, add a control group of non-specific IgG-nanoparticles to the cellular experiments. 

Discussion Please, add a discussion chapter to the manuscript according to the instructions to the authors of this journalIn particular, authors should discuss various drug delivery systems for inflammatory diseases, the dual role of PECAM-1 (proinflammatory and anti-inflammatory), and propose a mechanism of therapeutic efficacy of the developed drug delivery system. 

Сonclusions. Please, rephrase the conclusions. It is incorrect to write that «we designed antibodies» since anti-PECAM-1 antibodies were a gift from Wisconsin Blood Center (Wisconsin, USA). 

Minor English editing is required.

Round 2

Reviewer 3 Report

Authors made all necessary changes.